# First Genome Sequence of the Microcolonial Black Fungus *Saxispiralis lemnorum* MUM 23.14: Insights into the Unique Genomic Traits of the *Aeminiaceae* Family

**DOI:** 10.3390/microorganisms12010104

**Published:** 2024-01-04

**Authors:** Diana S. Paiva, Luís Fernandes, António Portugal, João Trovão

**Affiliations:** 1Centre for Functional Ecology (CFE)—Science for People & the Planet, Department of Life Sciences, University of Coimbra, Calçada Martim de Freitas, 3000-456 Coimbra, Portugaljtrovaosb@gmail.com (J.T.); 2FitoLab—Laboratory for Phytopathology, Instituto Pedro Nunes (IPN), Rua Pedro Nunes, 3030-199 Coimbra, Portugal; 3TERRA—Associate Laboratory for Sustainable Land Use and Ecosystem Services, Department of Life Sciences, University of Coimbra, Calçada Martim de Freitas, 3000-456 Coimbra, Portugal

**Keywords:** black meristematic fungi, comparative genomics, extremophiles, limestone, rock-inhabiting fungi, stress tolerance

## Abstract

*Saxispiralis lemnorum* MUM 23.14 is an extremotolerant microcolonial black fungus, originally isolated from a biodeteriorated limestone artwork in Portugal. This recently introduced species belongs to the *Aeminiaceae* family, representing the second member of this monophyletic clade. This fungus exhibits a unique set of characteristics, including xerophily, cold tolerance, high UV radiation tolerance, and an exceptional ability to thrive in NaCl concentrations of up to 30% while also enduring pH levels ranging from 5 to 11. To gain insights into its genomic traits associated with stress resistance mechanisms, specialization, and their potential implications in stone biodeterioration, we conducted a comprehensive genome sequencing and analysis. This draft genome not only marks the first for the *Saxispiralis* genus but also the second for the *Aeminiaceae* family. Furthermore, we performed two comparative genomic analyses: one focusing on the closest relative within the *Aeminiaceae* family, *Aeminium ludgeri*, and another encompassing the genome of different extremotolerant black fungi. In this study, we successfully achieved high genome completeness for *S. lemnorum* and confirmed its close phylogenetic relationship to *A. ludgeri*. Our findings revealed traits contributing to its extremophilic nature and provided insights into potential mechanisms contributing to stone biodeterioration. Many traits are common to both *Aeminiaceae* species and are shared with other black fungi, while numerous unique traits may be attributed to species-specific characteristics.

## 1. Introduction

*Saxispiralis lemnorum* MUM 23.14 is a microcolonial black fungus originally isolated from a deteriorated limestone funerary artwork at the Lemos Pantheon (Trofa do Vouga, Águeda, Portugal), a national monument of great architectural and artistic value and a remarkable example of the Portuguese stone-built cultural heritage [1]. This species was recently described through a comprehensive polyphasic approach, combining a multi-locus phylogeny with morphological, physiological, and ecological characterizations, and assigned to the *Aeminiaceae* family, *Mycosphaerellales* order, within *Capnodiales* s. lat., in the *Ascomycota* division [2]. The *Aeminiaceae* family was established by Trovão and colleagues [3] in 2019, and it currently comprises only two described genera, each with a single species, namely *Aeminium ludgeri* and *Saxispiralis lemnorum*. So far, this family solely hosts extremotolerant fungi characterized by slow growth and late melanization (occurring due to the very slow differentiation of heavy melanized arthroconidia), which have been found exclusively in association with various scenarios of deterioration affecting stone monuments in the Iberian Peninsula within the Mediterranean area [2,3].

Microcolonial black fungi, also referred to as rock-inhabiting or meristematic fungi, constitute an ecologically defined group of stress specialists [4]. Despite their diverse phylogenetic placement and distance, these fungi exhibit a set of convergent features that confer extensive stress tolerance, enabling them to thrive in hostile environments [5]. These unique morphological, physiological, and metabolic characteristics that allow their survival in various extreme environmental conditions encompass phenotypic plasticity, the ability to transition from a mycelial to a meristematic state, microcolonial growth, simple life cycles, and dispersal mechanisms reliant on vegetative fragmentation or poorly differentiated conidia-like cells [6,7,8,9,10,11,12]. Among these features, the most prominent and distinctive characteristic is the presence of strongly melanized cell walls, with melanin serving as a vital protective compound/pigment [11,12].

Although rocks present challenging niches, these fungi are commonly found in lithobiontic micro-communities, where they prevail under harsh and adverse conditions without being outcompeted by fast-growing fungi [8,13]. They are perfectly adapted to this extreme lifestyle and are among the most competent colonizers of this substrate, providing exceptional opportunities for species diversification [8]. When colonizing historical stone monuments, these fungi emerge as one of the most destructive microbial groups responsible for the deterioration and irreversible damage inflicted upon these assets, presenting three significant challenges: aesthetic, biophysical, and biochemical biodeterioration, and consequently posing a serious problem for the preservation of cultural heritage stone materials [14,15]. Despite their ecological significance, these fungi have been overlooked for an extended period, leaving their phenotypic traits and underlying mechanisms largely unexplored [16]. These poikilotolerant organisms have developed specific genetic adaptations and survival strategies to thrive in challenging habitats. Therefore, further genomic exploration is crucial for deciphering their stress resistance mechanisms and understanding how they cope with various stressors [17].

As for *Saxispiralis lemnorum* MUM 23.14, this peculiar rock-dwelling fungus exhibits a unique combination of traits, including xerophily, cold tolerance, and high tolerance to UV radiation, as well as an impressive ability to thrive in NaCl concentrations of up to 30% and to withstand pH levels ranging from 5 to 11 (the highest levels analyzed in both cases and thus, the range for these factors may still be greater than what has been identified—refer to [2] for more detailed information). Collectively, these characteristics enable this fungus to flourish under diverse physicochemical conditions and endure various stressors, highlighting its well-adapted survival strategy on stone surfaces. While the extremotolerant nature of this fungus is well established, both the genomic basis for such resistance and the putative biodeteriorative mechanisms have remained unexplored. Thus, to enhance our current understanding of the pathways through which this fungus adapts to extreme conditions and their potential contribution to stone biodeterioration, in this study, we generated and analyzed the first draft genome sequence of *Saxispiralis lemnorum* MUM 23.24. Additionally, we conducted two comparative phylogenomic analyses: one focusing on the closest relative within the *Aeminiaceae* family, *Aeminium ludgeri* DSM 106916, and the other encompassing genomes of other melanized fungi from extreme environments to investigate the evolutionary history, shared traits, and potential origins of these strains.

## 2. Materials and Methods

### 2.1. Fungal Strain Isolation, DNA Extraction, and Whole-Genome Sequencing

The *Saxispiralis lemnorum* MUM 23.14 isolate was obtained after the inoculation of a homogenized solution containing stone debris in 2 mL of sterile 0.9% (*w*/*v*) NaCl solution on Rose Bengal Agar Base (RB, Difco^TM^, Sparks, MD, USA) supplemented with streptomycin (0.5 g L^−1^) and incubated in the dark at 25 ± 2 °C for 6 months [2]. The emerging colony was subsequently transferred to an axenic culture and incubated in the dark at the same temperature for 60 days until full development and melanization of the cultures could be observed, prior to total DNA extraction.

Genomic DNA was extracted using the DNeasy Powerlyzer Powersoil Kit (Qiagen, Hilden, NRW, Germany), according to the manufacturer’s instructions at Genoinseq (Cantanhede, Portugal). DNA integrity was confirmed by electrophoresis on a 1% (*w*/*v*) agarose gel and quantified by the Qubit dsDNA High Sensitivity Assay Kit (Invitrogen Life Technologies, Waltham, MA, USA) on the Qubit 2.0 Fluorometer (Thermo Fisher Scientific, Waltham, MA, USA).

The DNA library for fungal genome sequencing was prepared from 0.5 nanograms of high-quality genomic DNA with the Nextera XT DNA Sample Preparation Kit (Illumina, San Diego, CA, USA) and sequenced using paired-end (PE) 2 × 150 bp on the NextSeq 550 Illumina^®^ platform at the Genome Medicine platform (IBIMED, University of Aveiro, Aveiro, Portugal). All procedures were performed according to the standard manufacturer’s protocols.

### 2.2. Genome Assembly, Annotation, and Functional Analysis

Genome assembly and annotation were conducted with the web-based Galaxy platform [18]. The initial reads were filtered with Trimmomatic v.0.38 [19] using the following parameters: ILLUMINACLIP, TruSeq3-PE, fa:2:30:10:8, SLIDINGWINDOW:5:25, and MINLEN:50. The obtained high-quality, adapter-free reads were then assembled with metaSPAdes v.3.15.4 [20] with default parameters. The putative presence of contaminants was screened and removed with kraken2 [21] and the RefSeq database. The initial assembly was somewhat fragmented (6281 sequences) and was thus further scaffolded with RagTag v.2.1.0 [22] and the *Aeminium ludgeri* DSM 106916 genome (GCA_004216415.1) [23]. Sequences with a size of <1000 bp were removed from the dataset. The final assembly metrics were then evaluated using Quast v.5.2.0 [24] and gfastats v.1.2.1. Genome completeness was estimated with Benchmarking Universal Single-Copy Orthologs (BUSCO) v.5.3.2 [25] using the ortholog dataset set for fungi (OrthoDB v.10) [26].

The software Barrnap v.0.9 [27] and ARAGORN v.1.2.36 [28] were used to identify genomic rRNA and tRNA genes, respectively. Complementarily, AUGUSTUS v.3.4.0 [29] was applied to perform coding gene predictions, and the species *Botrytis cinerea* selected for training. Functional annotation of coding genes in the assembly was conducted with DIAMOND v.2.0.15 [30] against the UniProtKB Swiss-Prot database (UniProt Consortium 2017) and Blast2Go v.1.2.14 [31,32], the EggNOG Mapper [33], and InterProScan [34,35], with all applications and default settings selected. The OmicsBox software v.2.0 was used to compile the functional annotation, the Interpro protein domains, families, and sites obtained, the Gene Ontology (GO) terms merged, and the GOSlim tool and the enzyme coding mapping tools applied. The predicted protein-encoding genes were also mapped to the Kyoto Encyclopedia of Genes and Genomes (KEGG) using BlastKOALA [36,37]. Moreover, carbohydrate-active enzymes (CAZymes/CAZome) were screened using the dbcan3 webserver [38], while biosynthetic gene clusters (BGCs) were screened with the antiSMASH web server v.6.0 [39], with the detection strictness set to strict.

### 2.3. Comparative Genomic Analysis

Two sets of comparative genomic analyses were conducted. The first analysis focused on the *Aeminiaceae* family using the *Aeminium ludgeri* DSM 106916 genome [23]. This species genomic data was also functionally annotated similarly to that described above and further compared with the genome obtained during this work using the OmicsBox software v.2.0. In addition, orthologous gene clusters for both genomes were annotated using the OrthoVenn3 web server [40]. For this analysis, the OrthoMCL [41] clustering algorithm was applied with default settings to compare and annotate ortholog groups. Gene Ontology (GO) terms for the biological process, molecular function, and cellular component categories, as well as enrichments, were obtained. In parallel, gene family contraction and expansion analysis were conducted with CAFE5 [42], as implemented in OrthoVenn3 web server. Whole-Genome Average Nucleotide Identity (ANI) comparisons between *Saxispiralis lemnorum* MUM 23.14 and *Aeminium ludgeri* DSM 106916 were also conducted using FastANI (https://github.com/ParBLiSS/FastANI, accessed on 10 October 2023). The second comparative genomic analysis was also performed using the OrthoVenn3 web server, but this time, it included other additional genomes of poikilotolerant black fungal species available from the NCBI, including *Cryomyces minteri* (NAJN01000001.1), *Friedmanniomyces endolithicus* (NAJP00000000.1), *Friedmanniomyces simplex* (NAJQ00000000.1), *Hortaea thailandica* (NAJL00000000), *Hortaea werneckii* (MUNK01000001.1), and *Rachicladosporium antarcticum* (NAJO00000000.1) [43], as described previously. All obtained data pertaining to the OrthoVenn3 analysis have been archived and can be accessed on Figshare via the following link: https://figshare.com/s/a88f1e03afc8eab5e20d (accessed on 11 December 2023).

## 3. Results and Discussion

In this study, we obtained a draft genome sequence of *Saxispiralis lennorum* MUM 23.14 and explored the functional genomic characteristics of this organism to shed light on its extremotolerance and potential biodeteriorative capabilities. The *Aeminiaceae* family currently represents a monophyletic clade, including only two genera and two species of extremotolerant black fungi, all of which have been exclusively detected in biodeteriorated stone monuments. Given that black fungi are prominent contributors to stone biodeterioration, it is imperative to gain a deeper understanding of their genomic traits to enable the development and improvement of tools aimed at protecting stone monuments from their deleterious effects.

### 3.1. Saxispiralis lemnorum MUM 23.14 Genome Assembly, Annotation, and Funtional Characterization

The *S. lemnorum* MUM 23.14 genome was assembled into 1414 scaffolds with a cumulative length of 25.84 Mbp and a 57.49% GC content. The scaffold N50 was 152,083 bp, with an average scaffold length of 18,278.01 bp and the largest scaffold being 744,002 bp. AUGUSTUS, Barrnap, and ARAGORN predicted the occurrence of 7957 genes, with three rRNAs (one 18S rRNA, one 28S rRNA, and one 5.8S rRNA) and forty-one tRNAs. The Benchmarking Universal Single-Copy Orthologs (BUSCO) completeness was estimated to be 90.2% for fungi (n = 758), with 684 complete BUSCOs, 684 complete and single-copy BUSCOs, 21 fragmented BUSCOs, and 53 missing BUSCOs (Figure 1). The Whole-Genome Shotgun project of *S. lennorum* MUM 23.14 has been deposited at DDBJ/ENA/GenBank under the accession JAWXCT000000000 (paper version JAWXCT010000000), linked to Bioproject PRJNA1021044 and biosample SAMN37543029.

From the overall genome annotation (Appendix A), the functional analysis considering GOslim results revealed that the top five most representative domains for (1) biological processes were as follows: anatomical structure development, signaling, cell differentiation, transmembrane transport, and lipid metabolic process; (2) cellular components were as follows: cytosol, plasma membrane, nucleus, mitochondrion, and nucleoplasm; and (3) molecular functions were as follows: hydrolase activity, transferase activity, catalytic activity acting on a protein, oxidoreductase activity, and DNA binding (Figure A1 and Appendix A).

On the other hand, out of the 7957 detected coding genes, InterProScan predicted information about protein function for 7308 that had an IPS assigned, and these could be further identified into 3626 protein families, 2599 protein domains, and 366 protein sites predicted. The top five most representative (1) families were as follows: (IPR027417) P-loop containing nucleoside triphosphate hydrolase, (IPR036291) NAD(P)-binding domain superfamily, (IPR036259) MFS transporter superfamily, (IPR015943) WD40/YVTN repeat-like-containing domain superfamily, and (IPR011009) protein kinase-like domain superfamily. (2) Domains were as follows: (IPR020846) major facilitator superfamily domain, (IPR000719) protein kinase domain, (IPR003593) AAA+ ATPase domain, (IPR001138) Zn(2)-C6 fungal-type DNA-binding domain, and (IPR001650) Helicase, C-terminal. (3) The sites were as follows: (IPR008271) serine/threonine-protein kinase active site, (IPR017441) protein kinase ATP binding site, (IPR019775) WD40 repeat conserved site, (IPR005829) sugar transporter conserved site, and (IPR020904) short-chain dehydrogenase/reductase conserved site (Figure A2 and Appendix A).

The OmicsBox enzyme coding mapping tool detected seven main enzyme classes, namely oxireductases, transferases, hydrolases, lyases, isomerases, ligases, and translocases. Among these, the most prevalent enzyme types found were hydrolases, transferases, and oxireductases (Figure A3 and Appendix A). Within these categories, the top five most abundant subclasses included enzymes transferring phosphorous-containing groups (transferases), enzymes acting on acid anhydrides (hydrolases), enzymes acting on ester bonds (hydrolases), enzymes acting on the CH-OH group of donors (oxireductases), and acyltransferases (transferases).

The Eggnog mapper revealed that the top five most relevant Clusters of Orthologous Genes (COGs) category groups were as follows: (S) unknown function (n = 1424), (G) carbohydrate metabolism and transport (n = 343), (O) post-translational modification (n = 338), (U) intracellular trafficking, secretion, and vesicular transport (n = 278), and (J) translation, ribosomal structure, and biogenesis (n = 272) (Figure A4 and Appendix A).

The dbCAN3 tool identified 488 carbohydrate-active enzymes (CAZome). In general, the frequency of the enzyme families was largely dominated by glycoside hydrolases (GH) and glycosyltransferases (GT), with a lower number of auxiliary activities (AA) and only a few carbohydrate-binding modules (CBM), carbohydrate esterases (CE), and polysaccharide lyases (PL) being detected (Figure A5 and Appendix A).

The fungal antiSMASH tool predicted six putative biosynthetic gene clusters (BGCs), namely: three T1PKS (type I polyketide synthase) clusters; one NRPS (non-ribosomal peptide synthetase) cluster; and two terpene clusters (Appendix A). While most of the predicted BGCs had somewhat low similarities with the MiBiG database [44,45], one T1PKS cluster stood out with a 100% match to 1,3,6,8-tetrahydroxynaphthalene, a critical precursor for DHN (1,8-Dihydroxynaphthalene) melanin biosynthesis. Additionally, another T1PKS cluster exhibited a 50% similarity to elsinochrome C, which is a perylenequinone, a class of polyketide-derived photosensitizers and dark-colored pigments owing to their abilities to absorb light energy, react with oxygen, and produce toxic reactive oxygen species (ROS) [46].

### 3.2. Aeminiaceae Comparative Genomic Analysis

Following a similar approach to that conducted for the *S. lemnorum* MUM 23.14, genome annotation was also attained for the *Aeminium ludgeri* DSM 106916, and the functional analysis details pertaining to this genome can be found in Appendix A and Figure A1, Figure A2, Figure A3, Figure A4 and Figure A5. Moreover, generic genomic comparisons can also be verified in Table 1.

The ANI score, which quantifies nucleotide-level genomic similarity within the coding regions of two genomes, was calculated to assess the similarities between the two *Aeminiaceae* species. Similarity is expected to be highest among individuals of the same species, whereas different species should share a lesser amount of genomic information. In this analysis, both species exhibited an 80.41% similarity, which aligns with their phylogenetic placement and distinct species status.

Overall, both genomes presented multiple similarities among themselves. The genome sizes of both species fall within the same size range, with the genome of *S. lemnorum* MUM 23.14 being slightly larger than that of *A. ludgeri* DSM 106916. In general, black fungi genomes typically range from 20 up to 50 Mbp [43], placing these representatives of the *Aeminiaceae* family on the lower end of the spectrum. The same trend is observed in the number of predicted genes, which also falls within the lower range [47]. This notable decrease in genome size and gene content could potentially be attributed to the process of adaptation to hostile environmental conditions [48]. Furthermore, while it is challenging to establish correlations between GC content and ecological characteristics in fungi (as discussed by Sterflinger et al., 2014 [49]), it is noteworthy that a high GC content appears to be a peculiarity of extremophilic black fungi. Interestingly, the genomes of the currently available *Aeminiaceae* representatives exhibit a higher GC content (57.49% for *S. lemnorum* MUM 23.14 and 58.57% for *A. ludgeri* DSM 106916) compared to the typically reported range for black fungi and yeasts (49–56.5%) [43,47]. Another interesting result is the significantly higher number of tRNAs predicted for *S. lemnorum* MUM 23.14, nearly double that of *A. ludgeri* DSM 106916. High amounts of tRNA reflect the significance of protein translation and degradation machinery in the cellular function of the fungus. Protein synthesis is a fundamental process for the survival and growth of any organism, and the abundant presence of these molecules indicates potential adaptation to various environments and specific translation requirements. Additionally, it reflects active maintenance of protein quality, essential for averting the buildup of defective proteins that could be detrimental to the cell [50]. Therefore, a high count of these molecules may indicate increased translation activity and an adaptive response to variable or stressful conditions.

Highly significant GOs and predicted proteomic features related to gene expression regulation, signal transduction, and transcription, cellular processes, and cell cycle control including division, proliferation, and apoptosis (as evidenced by the high abundance of WD40-repeat-containing domains, helicases, C-terminal domains, and protein kinase domains) were identified (Figure A1). The utmost represented protein domains were related to major facilitator transporters (MFS) (Figure A2), a type of secondary carrier responsible for transporting small solutes in response to chemiosmotic ion gradients [51]. MFS are of particular relevance since they mediate resistance to toxic compounds and provide genomic evidence of the species extremotolerance (and potentially even to restoration/protection treatments) [47]. Simultaneously, high levels of the predicted IPR001138 (Zn (2)-C6 fungal-type DNA-binding domain) and the presence of IPR007219 (transcription factor domain, fungi) have been previously reported in the halotolerant *Hortaea werneckii* and the Antarctic *Friedmanniomyces endolithicus*. Both domains act on various cellular and metabolic processes, likely enabling metabolic system functionality under unfavorable and extreme conditions [43].

Regarding the enzymatic characteristics, the conducted analysis highlighted a very representative level of hydrolases, transferases, and oxireductases (Figure A3). The high abundance of hydrolases and oxireductases is relevant considering that, while they are required for lignin and cellulose degradation [52,53], they were also found to be related to enzymes needed for black slate degradation in the basidyomycete *Schizophyllum commune* [54]. This may potentially serve as an indicator or be linked to the deteriorative potential of these species. Complementarily, CAZymes, responsible for carbohydrate and glycoconjugate biosynthesis, modification, and degradation [55], were found in similar numbers in both species, with minimal variation (Figure A5). Glycoside hydrolases (GHs) and glycosyl transferases (GTs) were the most abundant gene families, consistent with findings in other black fungi [43,47]. In particular, CAZyme families GH5, GH3, GH16, GH31, GH43, and GH47 (glycoside hydrolases) were found to be enriched in these two species, as well as GT1, GT2, GT22, GT31, GT4, and GT90 (glycosyl transferases) (Figure A6). Curiously, families GT22 and GH5 were also found to be enriched in the extremotolerant species *F. endolithicus* and *H. werneckii*, whereas only GH47 was enriched in *H. werneckii* [43]. In contrast, auxiliary activities (AAs), carbohydrate-binding modules (CBMs), and carbohydrate esterases (CEs) were present in much lower abundance. Polysaccharide lyases (PLs) exhibited noticeable depletions, with only a few being detected (specifically PL1, PL3, PL4, and PL26). In general, many black yeasts and fungi lack various pectinases, which are typically associated with the degradation of plant polysaccharides, a feature that is often considered indicative of a more generalist lifestyle [47,56].

Fungi are known to produce a wide variety of secondary metabolites (bioactive molecules not required for growth or other essential processes), which play a crucial role in their ecological characteristics and potential interactions [57]. These bioactive molecules are typically the result of biosynthetic gene clusters (BGCs) composed of enzymes, regulators, and transporters contributing to a biosynthetic pathway. They are categorized as terpene synthases (TS), polyketide synthetases (PKS), nonribosomal peptide synthetases (NRPS), and dimethylallyl tryptophan synthases (DMAT) [58]. When considering the results obtained for the predicted BGCs, it was observed that several clusters in *S. lemnorum* MUM 23.14 and *A. ludgeri* DSM 106916 exhibited low similarities with potential known homologs. Therefore, it is expected that they encode unique and unknown compounds. On the other hand, the most notable result regarding the BGCs concerns the T1PKS present in both *S. lemnorum* MUM 23.14 and *A. ludgeri* DSM 106916, showing high similarity to various melanin clusters. Unlike other filamentous fungi, black yeasts lack the genetic component arrangement in biosynthetic clusters for melanin synthesis [47]. Therefore, concerning melanin synthesis, *Aeminiaceae* species are naturally more closely related to filamentous black fungi [59].

#### 3.2.1. Comparative KEGG Analysis and Stress-Related Genes

In our analysis, GhostKoala annotated 3065 genes, accounting for 38.5% of the total gene content in *S. lemnorum* MUM 23.14 and 3174 genes in *A. ludgeri* DSM 106916, which corresponds to 39.2%, while the KEGG analysis unveiled several distinctive genomic characteristics (Table A1). The results obtained reveal that both species are methylotrophs, assimilating formaldehyde through the xylulose monophosphate pathway (or dihydroxyacetone cycle)—methane metabolism. Additionally, the signature modules indicate that both species can assimilate nitrate through the assimilatory nitrate reduction into ammonia, nitrogen metabolism, and can also perform assimilatory sulfate reduction, converting sulfate to sulfide (H_2_S)—sulfur metabolism. Notably, *A. ludgeri* DSM 106916 possesses an additional capability for ATP synthesis involving the NADH dehydrogenase (ubiquinone) Fe-S protein/flavoprotein complex in the mitochondria. Methylotrophic organisms utilize single-carbon compounds, including methane, methanol, formate, and carbon monoxide, as their carbon source for growth [60]. In fact, various black fungal species are known to thrive on unconventional carbon sources. Some authors have suggested that ionizing radiation, alternative carbon fixation pathways, and the hypothetical use of photochemical energy to reduce CO_2_ are potential mechanisms for these unique organisms to obtain chemical energy [61,62,63,64]. While additional research is needed, the ability for methylotrophic metabolism could confer a significant advantage, particularly in oligotrophic and extremophilic environments where these organisms can thrive under nutrient-poor conditions or by utilizing alternative and unconventional energy sources [63]. Additionally, nitrate assimilation may contribute to CaCO_3_ mineralization by increasing carbonate alkalinity, while biological sources of H_2_S can play a role in the biodeterioration of concrete and inorganic pigments [65,66,67].

Both species can synthesize farnesal and trans-farnesol. However, the complete pathway for metabolizing products from the terpenoid backbone biosynthesis to generate carotenoids, such as neurosporaxanthin, γ-carotene, and β-carotene, was only identified in *A. ludgeri* DSM 106916. Surprisingly, despite the visual prominence of these compounds in young colonies of *S. lemnorum* MUM 23.14, this analysis suggests that the carotenoid biosynthesis pathway in this species may either be incomplete or that the relevant components have remained undetected or are uncharacterized. The ability to synthesize carotenoids highlights a crucial extremotolerant trait, as various mycosporines and carotenoids are recognized for providing protection against UV radiation in different extremophilic fungal species [6,8,9,11,68,69,70]. Typically, carotenoids in black fungal species are in the cell walls, usually concealed by melanin (only visible in melanin mutant strains) [71,72]. Nonetheless, the unique development of the *Aeminiaceae* family representatives provides the uncommon opportunity to clearly detect the presence of carotenoids. This is evident as colonies initially appear in a vibrant orange color for *S. lemnorum* MUM 23.14 and a pinkish hue for *A. ludgeri* DSM 106916, eventually darkening due to the slow differentiation of heavily melanized arthroconidia (refer to [2,3] for more detailed information). Furthermore, another highly significant group of pigments are melanins. Our analysis confirmed that both species possess the genetic machinery for synthesizing both eumelanin and pheomelanin via the DOPA-melanin pathway, as well as pyomelanin (allomelanin) through the L-tyrosine degradation pathway. Fungal melanins are hydrophobic pigment biopolymers formed by the oxidative polymerization of phenolic or indolic compounds. Usually, melanins appear brown to black, although variations can result in other colors, including green. Melanins serve as versatile protectors against various stressors, providing cellular defense against UV radiation, oxidative agents, and extreme temperatures [73,74,75].

Furthermore, the KEGG pathway analysis also demonstrated that both species have the ability to synthesize trehalose through the starch and sucrose metabolism pathway. Trehalose is known for stabilizing both enzymes and membranes, and it plays a pivotal role in enabling anhydrobiotic organisms to thrive in low-water-activity environments and even endure complete dehydration [76]. Also, both possess the mechanisms to synthesize trehalose-6-phosphate (T6P), which has been linked to diverse roles, such as serving as an energy source and acting as a protectant against stressors like heat, freezing, starvation, dehydration, and desiccation [47]. At the same time, both species are capable of glycerol production. In black meristematic fungi, trehalose accumulates in response to high temperatures, while intracellular glycerol regulates osmotic potential under NaCl stress [76]. Glycerol accumulation is also a response to cold temperatures and functions as a cryoprotectant in fungi [77]. Consequently, both compounds may play crucial roles in imparting an extremotolerant profile to these species.

When analyzing the KEGG pathways related to environmental information processing, the two-component system revealed slight differences in responses to hyperosmotic stress in the two fungal species. *A. ludgeri* DSM 106916 exhibited the involvement of sensor histidine kinase SLN1 and NIK1, the phosphorelay intermediate protein YPD1, and the response regulator SKN7, while in *S. lemnorum* MUM 23.14, only the sensor histidine kinase SLN1 and NIK1 and the response regulator SKN7 were detected, with YPD1 missing. Moreover, when analyzing the MAPK signaling pathway for yeasts, it becomes evident that both species share highly similar pathways in response to cell wall stress, high osmolarity, and starvation.

In addition, our genome annotations for both *Aeminiaceae* species revealed numerous genes responsible for proteins associated with environmental stress responses. These proteins are recognized for aiding fungi in coping with a range of stresses, including oxidative, heat, acidic, osmotic, and UVA stress [72,78]. The identified genes encompassed several superoxide dismutases, catalases, mitogen-activated protein kinases (Hog1), heat shock proteins, calcium-transporting ATPases, alpha-1,2/1,3/1,6-mannosyltransferases, adenosyl homocysteinases, and photolyases (Table A2). The presence of superoxide dismutases is particularly significant, as their involvement in superoxide catalysis serves as a crucial marker of resistance to oxidative and salt stress [79]. Furthermore, several alcohol dehydrogenases (Adh) were also detected. A physiological role of Adh has been reported in many biochemical pathways, including stress tolerance, pathogenicity, detoxification, and substrate specificity [47]. While the presence of these coding genes may potentially contribute to the extremophilic nature of these organisms, it is important to note that their regulatory and expression patterns can also have a profound impact [80,81]. For this reason, further studies on environmental stress resistance are necessary, particularly considering that, thus far, they have been sparsely explored for these types of fungi.

Concerning melanin biosynthesis, in addition to the previously mentioned genetic framework for synthesizing both eumelanin and pheomelanin through the DOPA-melanin pathway, as well as pyomelanin via the L-tyrosine degradation pathway, the predicted BGCs obtained from the genome analysis also revealed the complete cluster involved in the DHN-melanin pathway synthesis with a 100% match. This indicates that, similar to certain black yeasts and fungi [47,72], both *Aeminiaceae* species possess the genetic architecture for synthesizing all types of melanin.

As observed in *Fonsecaea monophora*, both *Aeminiaceae* species possess genes encoding APSES and STE transcription factors (e.g., stuA and STE50), responsible for regulating yeast-to-hyphae transitions, while only *A. ludgeri* DSM 106916 contains the *Opsin-1* gene, a conserved UV-A light response light sensor found in the carotenoid biosynthetic and cryptochrome-photolyase pathways [72,82,83,84].

Finally, two carbonic anhydrases and seven cysteine synthases were detected in *S. lemnorum* MUM 23.14, whereas *A. ludgeri* DSM 106916 possessed only two of each, genes known to be upregulated in *Aspergillus niger* during the weathering of potassium-bearing minerals [85].

#### 3.2.2. Comparative OrthoVenn3 Analysis

A comparative analysis using predicted orthologous gene clusters was conducted with the OrthoVenn3 web server. Orthologs are genes in different species that have evolved from a common ancestral gene through speciation and often maintain their original function throughout evolution. Comparing orthologs plays a pivotal role in comparative genomic studies, as it enables the assessment of evolutionary relationships across genome structure, gene function, and taxonomic classification among different species [86]. Both *Aeminiaceae* species formed 5643 clusters, with three overlaps (when one or more members of a cluster are shared by different clusters), 5496 single-copy clusters (single-copy genes in each species), 16,060 proteins, and 4668 singletons (29.07%) (without orthologs) between the genomes. Out of these, 5570 conserved orthologous clusters were shared amongst both genomes, with only 21 and 52 being uniquely identified in *S. lemnorum* MUM 23.14 and *A. ludgeri* DSM 106916, respectively (Figure 2). The greater number of shared clusters between these species underscores their close genetic relationship, whereas the varying number of unique clusters may be associated with traits specific to each genus.

Within the 5570 shared clusters, there was a combined total of 11,233 proteins, with each species contributing approximately 50% of these proteins. Most proteins within these clusters were found to be involved in basic biological (GO:0008150, 17.56%), metabolic (GO:0008152, 17.25%), and cellular processes (GO:0009987, 10.47%), as well as in cellular (GO:0044237, 12.43%) and macromolecule (GO:0043170, 10.60%) metabolic activities. Molecular functions were primarily related to oxidoreductase activity (GO:0016491, 21.04%) and hydrolase activity (GO:0016787, 14.63%), whereas cellular components were mainly associated with membrane (GO:0016020, 21.63%) and cellular components (GO:0005575, 17.55%) (Appendix A).

Among the clusters shared by both *Aeminiaceae* species, several noteworthy orthologs were recognized, including the following: DNA repair (GO:0006281, clusters 70, 78, 1182, 1222, 1239, 2189, 2622, 2933, 3283, 3833, 3963, 4787, 4820, and 4972); single-species biofilm formation on inanimate substrate (GO:0044011, clusters 856, 1134, 2563, 2800, 4142, 4461, and 5429); stress-activated protein kinase signaling cascade (GO:0031098, clusters 130, 176, 570, 1569, 1933, 2179, 2190, 2753, 2870, and 5241); stress-activated MAPK cascade (GO:0051403, cluster 2072); pigment biosynthetic process (GO:0046148, cluster 704); melanin biosynthetic process (GO:0042438, cluster 199); carotenoid biosynthetic process (GO:0016117, cluster 5558); responses to UV (GO:0009411, clusters 607, 1763, 2213, and 2970); UV-damage excision repair (GO:0070914, clusters 672, 2035,2456, 3921, 5041, 5472, and 5519); response to oxidative stress (GO:0006979, clusters 165,523,1482,2755, 2781, 2929, 3127, 3500, 3680, and 4769); cellular response to oxidative stress (GO:0034599, clusters 625, 940, and 3601); regulation of mRNA stability involved in response to oxidative stress (GO:2000815, cluster 2306); trehalose metabolism in response to stress (GO:0070413, clusters 3428, 3982, and 5260); trehalose catabolic process involved in cellular response to stress (GO:1903134, cluster 2199); responses to heat (GO:0009408, clusters 565, 2743, 3002, 3103, 3544, 3628, 4055, 5098, 5088, and 5490); response to salt stress (GO:0009651, cluster5234); cellular response to salt stress (GO:0071472, cluster 1120); response to osmotic stress (GO:0006970, cluster 1702); cellular response to osmotic stress (GO:0071470, cluster 2319); response to acidic pH (GO:0010447, clusters 2847 and 3000); response to radiation (GO:0009314, clusters 1104 and 2225), response to X-ray (GO:0010165, cluster 3665), and response to toxic substance (GO:0009636, cluster 1713). These processes are potentially linked with pathways involved in responding to environmental stressors, thereby confirming the exceptional poly-extremotolerance of these fungi, allowing them to survive a wide range of hostile conditions.

Notably, we also discovered numerous clusters related to pathogenesis (GO:0009405, clusters 9, 724, 866, 915, 957, 1422, 1477, 1610, 1669, 1678, 2080, 2141, 2146, 2157, 2181, 2535, 2645, 2724, 2728, 2888, 2937, 3132, 3329, 3366, 3459, 3633, 3736, 3834, 3838, 4570, 4614, 4688, 4912, 4970, 5338, and 5628). Many of the pathogenicity-related proteins are homologous to species of opportunistic human fungi, such as *Trichophyton rubrum* and *T. mentagrophytes* (dermatophyte fungi that colonize the human skin surface and are the most common causes of conditions like athlete’s foot, onychomycosis, and dermatophytosis), *Microsporum canis* (the agent causing tinea capitis), *Aspergillus fumigatus*, and *Alternaria alternata* (associated with different infections and allergic reactions), along with opportunistic yeasts such as *Candida albicans* (implicated in various human infections). This finding raises the question of whether these fungi could potentially be opportunistic for humans and pose health-related issues. Furthermore, mycotoxin biosynthesis (GO:0043386, cluster 4145) and aflatoxin biosynthetic processes (GO:0045122, clusters 1893 and 4864) were also identified. Aflatoxins, a group of polyketide-derived furanocoumarins known for their extreme toxicity and carcinogenic properties, have so far not been linked to black fungi.

When considering unique clusters for each species, oxidoreductase activity acting on the CH-OH group of donors (GO:0016614) and zinc ion binding (GO:0008270) were enriched by a 2-fold count in *S. lemnorum* MUM 23.14 genome (Appendix A), while molybdate ion transmembrane transporter activity (GO:001509) and the biological process of response to xenobiotic stimulus (GO:0009410) were enriched by a 2-fold count in the *A. ludgeri* DSM 106916 genome (Appendix A). Studies have shown a correlation between the maximum tolerated salinity in certain halophilic and halotolerant fungi and the number of genes involved in the cellular oxidative stress response [87]. This suggests a potential link between the antioxidant capacity of cells and their ability to thrive in high-salinity environments. Additionally, the presence of cell wall pigmentation is believed to play a crucial role in extremotolerance by serving as a protective barrier against oxidative damage. As previously mentioned in Section 3.2.1, melanin, a well-known component in the survival of melanized fungi, provides protection under extreme conditions, including exposure to high or low temperatures, extreme pH levels, or osmotic stress [43]. Also, among the clusters exclusive to each species, certain proteins related to X-ray radiation responses (GO:0010165, cluster 73) and aflatoxin biosynthetic process (GO:0045122, cluster 75) were exclusively found in *S. lemnorum* MUM 23.14. Previous studies have demonstrated that melanized fungi can withstand various types of radiation exposure, including ultraviolet (UV), X-rays, gamma radiation, nuclear radiation, and others, without significantly impacting their vitality and metabolic activity. The authors have suggested that this ability might be an evolutionary adaptation to repair DNA damage induced by extreme conditions [43]. As for genomic features unique to *A. ludgeri* DSM 106916, they included sporulation, resulting in the formation of a cellular spore (GO:0030435) related to the *LaeA* gene.

Using single-copy genes to infer the evolutionary history of the *Aeminiaceae* family revealed that the divergence between *S. lemnorum* MUM 23.14 and *A. ludgeri* DSM 106916 was prominently marked by gene family contraction events. This dynamic process of gene family expansion and contraction, following their divergence, has led to noticeable differences in their genome sizes and gene content. Furthermore, 1 expansion and 48 contractions were detected in the *S. lemnorum* MUM 23.14 genome, while 3 expansions and 19 contractions were detected in the *A. ludgeri* DSM 106916 genome (Appendix A). For both species, the clusters that were lost include some related to spore formation, which may be somehow linked to the peculiar sporulation process exhibited by these species.

### 3.3. Additional Poikilotolerant Black Fungal Species Comparative Genomic Analysis

In the comparative genomic analysis with additional poikilotolerant black fungal species using the OrthoVenn3 web server and predicted orthologous gene clusters (Figure 3A), we identified 12,871 orthologous clusters encoding 104,992 proteins. Among these clusters, 3 were single-copy and 234 showed overlaps. Furthermore, approximately 11.45% of these clusters were singletons, meaning they lacked orthologs in the other species. Of these, 3637 conserved orthologous clusters were shared amongst all genomes. The total number of gene clusters across the eight species ranged from 5777 to 8651, with *S. lemnorum* MUM 23.14 having the lowest count and *Friedmanniomyces simplex* having the highest. Notably, *S. lemnorum* MUM 23.14, *F. simplex*, and *Cryomyces minteri* exhibited the highest number of singletons. The abundance of singletons in these species potentially suggests an increased presence of unique genes associated with species-specific characteristics. However, it is essential to thoroughly scrutinize these singletons to determine whether they are genuinely unique or could potentially be sequencing artifacts or assembly errors.

Examining the 3637 clusters shared among all eight analyzed black fungal species, a total of 44,226 proteins were identified. Notably, both *Aeminiaceae* species, along with *Hortaea thailandica*, exhibited the smallest contributions, each contributing approximately 8%. In the cluster list, GO:0055085, related to transmembrane transport, stood out with the highest number of associated proteins (60-fold count). Within these shared clusters, most proteins were associated with biological processes (GO:0008150, 17.56%), while molecular functions were predominantly linked to oxidoreductase activity (GO:0016491, 18.94%), and cellular components were mainly associated with the membrane (GO:0016020, 18.72%).

A higher number of clusters were shared between the two *Aeminiaceae* species, which reflects their closer genetic relationship, with 325 clusters being uniquely detected for these two species (Appendix A). Among these, there was a significant enrichment (9-fold) of GO:0009405, which is related to pathogenesis. On the other hand, 17 exclusive clusters were identified in *S. lemnorum* MUM 23.14 (involving 40 proteins) and 18 in *A. ludgeri* DSM 106916 (with 38 proteins). The functions linked to these specific clusters remain unknown or poorly characterized.

To better understand the impact of evolution on key regulatory gene families over time and to clarify the evolutionary relationships among the eight black fungal species analyzed, we explored the events of gene family expansion and contraction (Figure 3B). Interestingly, the results revealed that gene family contraction events were more common than gene family expansion events in the evolutionary history of the species under study. Specifically, the *Aeminiaceae* lineage exhibited a remarkable pattern of contractions, surpassing those noted in other lineages. Particularly striking was a significant contraction event that led to the loss of 26 clusters, constituting one of the most extensive contraction events in the dataset. This large contraction observed in *Aeminiaceae* is particularly marked by a 60-fold enrichment of genes related to transmembrane transport (Appendix A). Another significant event comes to light when we analyze the divergence times. The observed split within *Mycosphaerellales*, which resulted in the divergence of the *Aeminiaceae* and *Teratosphaeriaceae* (encompassing both *Hortaea* species) families, seems to have some connection with the substrates from which these species are isolated, as *Aeminiaceae* has been exclusively found on stone surfaces, while *Hortaea* colonizes a broader range of substrates. Also, when we consider the separation of the clade containing both *Aeminiaceae* and *Hortaea* species from the other analyzed black fungi, temperature appears to be the most significant factor.

Furthermore, 2 expansions and 25 contractions were identified in the *S. lemnorum* MUM 23.14 genome, while 8 expansions and 11 contractions were detected in the *A. ludgeri* DSM 106916 genome (Appendix A) for this analysis. In the case of *S. lemnorum* MUM 23.14, contractions were notably associated with transmembrane transport (GO:0055085) (64-fold count), whereas expansions were enriched in transferase activity (GO:0016740) (4-fold count). Conversely, for *A. ludgeri* DSM 106916, contractions were particularly related to transcription DNA-templated (GO:0006351) (7-fold count), while expansions were significantly enriched in transmembrane transport (GO:0055085) (59-fold count).

Considering these results, this study undeniably demonstrated the functional potential of both species within the *Aeminiaceae* family, establishing a link between their demonstrated extremotolerance and their potential roles in biodeterioration resulting from their proliferation. However, it is essential to emphasize that while genomic analysis provides fundamental insights into an organism’s capabilities, it must be complemented by integrating data from genomics, transcriptomics, proteomics, and metabolomics. Such a comprehensive approach offers a deeper understanding of gene functions, their expression patterns, and the environmental conditions influencing them. The integrated analysis of multi-“omics” data enables scientists to bridge genomic information with the real-life functions of organisms in their natural habitats. Furthermore, the overall assembly quality for *A. ludgeri* DSM 106916 surpasses that of *S. lemnorum* MUM 23.14. Despite their phylogenetic proximity and the demonstrated similarities in this study, thus far, physiological analyses conducted on both species have revealed that *S. lemnorum* MUM 23.14 exhibits a significantly higher capacity for tolerating a broader range of conditions [2,3]. As a result, the undetermined parts of the genome may still conceal significant and distinctive insights.

## 4. Conclusions

Microcolonial black fungi, a group of ascomycetes known for their high stress tolerance, meristematic growth, and constitutive melanin formation, are resilient inhabitants of harsh environments. These fungi stand as compelling models for studying extreme phenotypes across various scientific domains, including ecology, astrobiology, clinical research, and material sciences. Their extraordinary resilience continues to pique scientific curiosity, offering valuable insights into life’s tenacity in extreme conditions and its potential applications. This study provided insights into the unique genomic characteristics of *S. lemnorum* MUM 23.14, revealing its extremotolerance traits and allowing us to anticipate the potential biodeteriorative impact of its proliferation on stone monuments. Furthermore, the shared gene families and pathways with *A. ludgeri* DSM 106916 underscore their close genetic relationship, contributing to a deeper understanding of the unique nature of the *Aeminiaceae* family. Additionally, the abundance of conserved orthologous clusters with various other black fungal species further enhances our comprehension of their evolutionary relationships. Therefore, similar research efforts hold the potential to yield valuable insights that can guide the development of more effective conservation practices when microcolonial fungi are observed colonizing invaluable cultural heritage monuments.

## Figures and Tables

**Figure 1 microorganisms-12-00104-f001:**
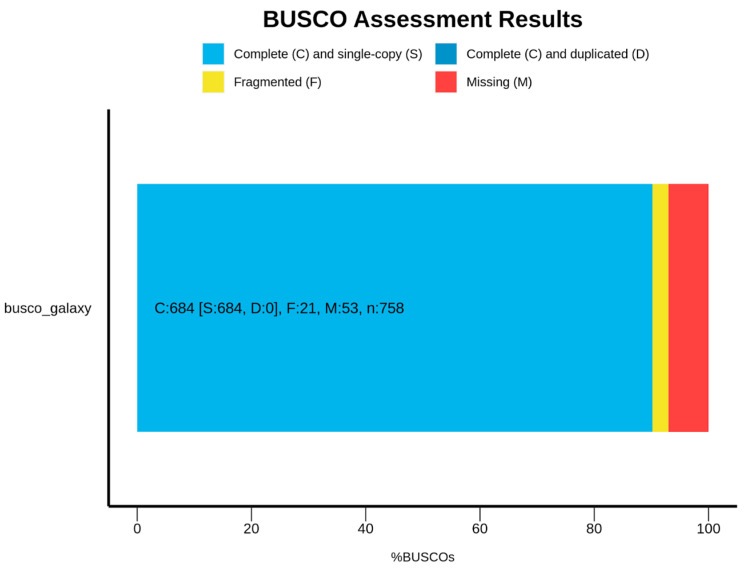
Genome completeness as evaluated with BUSCO.

**Figure 2 microorganisms-12-00104-f002:**
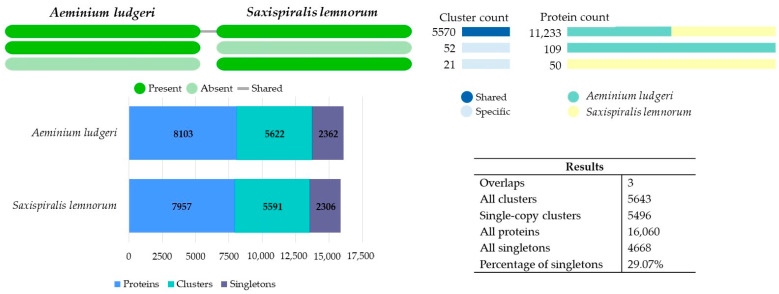
OrthoVenn3 web server results comparing the genomes of *Saxispiralis lemnorum* MUM 23.14 and *Aeminium ludgeri* DSM 106916, depicted in a diagram showing the distribution of shared and unique clusters of orthologous groups between the two *Aeminiaceae* species.

**Figure 3 microorganisms-12-00104-f003:**
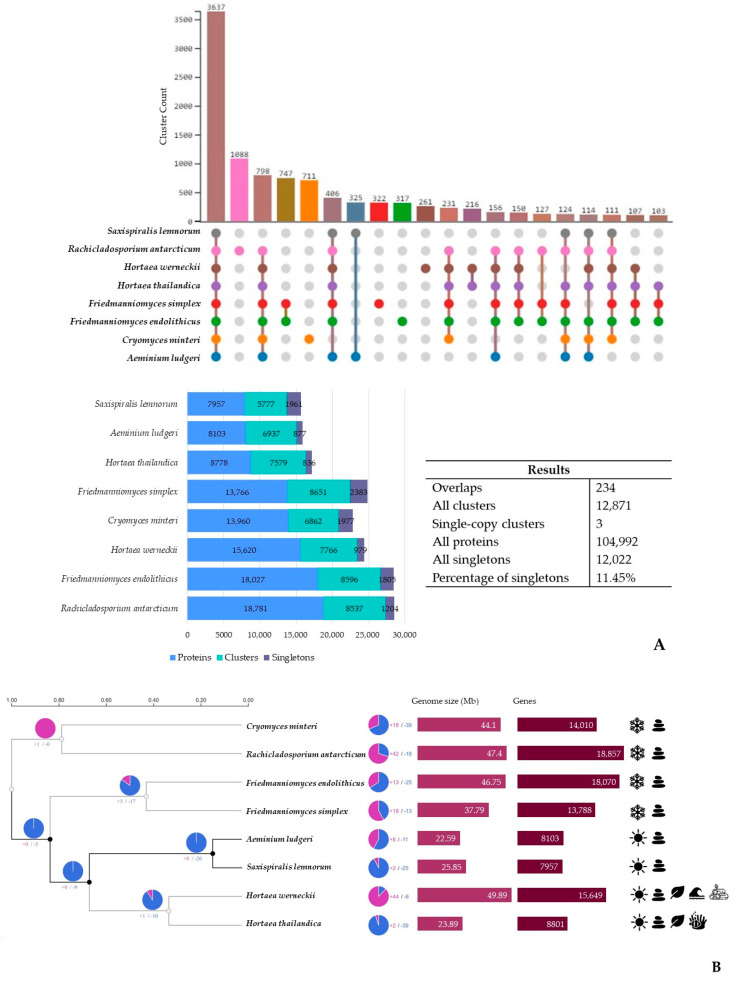
Results from the OrthoVenn3 web server, comparing the genomes of the two *Aeminiaceae* representatives with six other poikilotolerant black fungal species. (**A**) Diagram illustrating the distribution of shared and unique clusters of orthologous groups between the eight species. (**B**) Phylogenetic tree depicts the evolutionary timeline of the species, while the pie chart illustrates the number of gene families that have expanded (in purple) or contracted (in blue) during evolution, with genome size, gene numbers and isolation source also displayed for each compared species on the right (
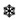
 Antarctic climate; 
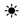
 moderate climate; 
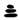
 rock; 
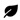
 plant; 
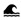
 water; 
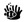
 coral; 
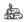
 wood).

**Table 1 microorganisms-12-00104-t001:** General genomic comparisons between *Saxispiralis lemnorum* MUM 23.14 and *Aeminium ludgeri* DSM 106916 genomes.

Data	*S. lemnorum* MUM 23.14	*A. ludgeri* DSM 106916
Size	25,845,107 bp	22,585,551 bp
GC content	57.49%	58.57%
Scaffold number	1414	228
Scaffold N50	152,083 bp	181,365 bp
Genes	7957	8103
rRNAs	3	2
tRNAs	41	23
CAZome	488	581
BGCs	6	11

## Data Availability

All relevant data are presented in the paper. The Whole-Genome Shotgun project of *Saxispiralis lemnorum* MUM 23–24 have been deposited at DDBJ/ENA/GenBank under the accession JAWXCT000000000 (paper version JAWXCT010000000), linked to Bioproject PRJNA1021044 and biosample SAMN37543029. All obtained data pertaining to the OrthoVenn3 analysis have been archived and can be accessed on Figshare via the following link: https://figshare.com/s/a88f1e03afc8eab5e20d (accessed on 11 December 2023).

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
