# Peer review of "First Genome Sequence of the Microcolonial Black Fungus Saxispiralis lemnorum MUM 23.14: Insights into the Unique Genomic Traits of the Aeminiaceae Family"

_microorganisms, 2024, doi:10.3390/microorganisms12010104_

Round 1

Reviewer 1 Report

Comments and Suggestions for Authors

A paper titled "First Genome Sequence of the Microcolonial Black Fungus Saxispiralis lemnorum: Insights into the Unique Genomic Traits of the Aeminiaceae Family" has been submitted to Microorganisms by Diana S. Paiva and coworkers in the team of João Trovão. This paper is basically a description of the genome of this stone inhabiting fungus and an attempt to correlate this genome with the particular physiological features related to life on the rocks.

One point that is not discussed is why will sometimes rocks be colonized by this Saxispiralis fungus, some other times by bacteria such as Modestobacter or Geodermatophilus and some other times remain naked ?

One missing element is a deposit in two strain collections to ensure other researchers can reproduce the results described here.

I also tried to retrieve the genome sequence on the NCBI using accession number JAWXCT000000000 but I failed. The accession should be available publicly.

I consider the paper could focus more on the particular traits related to deterioration of stones, what are its sources of carbon and energy, and what is it that it secretes that deteriorate stone. For the first point, I expected some assays on ability to grow on volatile carbon sources and a discussion on the genes present. For the second, the capacity to secrete acids.

The paper is otherwise well written, contains a lot of information and deserves to be published provided some minor modifications are made.

Minor corrections

-The language is in general good but is some instances a bit too precious. For instance, what is the difference between "poikilotolerant" and "xerophily" used in the introduction. Also "anhydrobio...", is it any different from the other two ?

-The eggnog map (Fig A4) should not contain categories "S" and "others" that dwarf the others and only underline the limits of this approach.

-There is no physiology at all. I would have liked a UV resistance assay to show the fungus is indeed adapted to its habitat.

-There is no morphology at all. I would have liked a photo of the in situ colonies on stone surfaces, of in vitro colonies, of micrograph to show the hyphae and sporangia.

Comments on the Quality of English Language

good in general, minor errors

Author Response

The authors would like to express sincere appreciation for your valuable feedback on our manuscript. Please see the attachment for a point-by-point response to the comments.

Reviewer 2 Report

Comments and Suggestions for Authors

The manuscript  "First Genome Sequence of the Microcolonial Black Fungus Saxispiralis lemnorum: Insights into the Unique Genomic Traits of the Aeminiaceae Family" by Paiva et al mainly studied the draft genome sequence of Saxispiralis lemnorum MUM 23.14 and illustrated the genomic traits associated with its stress resistance mechanisms. Overall, the work is very significant and, but several concerns (mainly with poor representation and unavailability of the raw datasets) in the manuscript that need to be addressed for further processing of this manuscript.

1. Please include the strain name (MUM 23.14) in the title.

2. 24-25: What are these distinctive features? Is this distinctive from Aeminium ludgeri or extremo-tolerant black fungi.
3. Please write the concluding statement/major findings in the abstract.

4. Why do you compaired the test strain with Aeminium ludgeri DSM 106916 genome? Is Aeminium ludgeri extreme-tolerant or susceptible? Do they deteriorate the stone monuments?
It is important to compare the test genome with the closely related genome (a known extreme-tolerant and a susceptible strains) so that genomic clues behinds the novel feature of Saxispiralis lemnorum can be disclose. You may follow the published papers https://doi.org/10.3389/fbioe.2023.1257705.

5. Why "Aeminium ludgeri DSM 106916" is used in the assembly? Please add the reason.

6. Figure 1 can be place in Supplementary

7. The sequence should be publicklly available in the NCBI. I could not find the raw data using the accession JAWXCT000000000 or PRJNA1021044 or SAMN37543029.

8. The result and discussion should focus on the important/ novel properties  associated with its stress resistance mechanisms, not the large/common description of the genomic properties which can be found in NCBI. Please rewrite this section.

Author Response

(The authors gave the same response as above.)

Reviewer 3 Report

Comments and Suggestions for Authors

The manuscript is interesting and well-structured. Microcolonial black fungi are known for their high stress tolerance, meristematic growth and melanin formation. These fungi can serve as compelling models for studying extreme phenotypes in a variety of scientific fields, including ecology, astrobiology, and materials science. This is an important and useful study of unique genomic characteristics of S. lemnorum MUM 23.14, reveal its extremotolerance traits and allow to anticipate the potential biodeteriorative impact  on stone monuments.

Author Response

We thank the reviewer for this encouraging comment, and we are very grateful for the overall positive appreciation of the manuscript. We truly appreciate the time invested, insightful feedback provided, and the contribution made to the success of the manuscript.

Round 2

Reviewer 2 Report

Comments and Suggestions for Authors

Major comments

1. The conclusions of this study cannot be drawn with the existing comparative analysis. It is likely that the unique genomic features (associated with the extremo-tolerance traits) of S. lemnorum MUM 23.14 are in fact common features of most fungal species associated with their survival and growth in natural environments. The comparative analysis must contain closely related non-extremophilic species to identify the unique genes/or gene clusters associated with the novel (extremophilic) characteristics of S. lemnorum MUM 23.14.

2. Raw and annotated sequences should be publicly available so that the reviewer can check and assess the findings.

3. Result and discussion need to rewrite with focusing only the major findings. Several sections (for eg, 450-478) need to represent as tables or supplementary materials. 

Other comment's

260-269:the comparison is not appropriate since both  S. lemnorum MUM 23.14 and A. ludgeri DSM 106916 are extremophilic. Please compare tRNA count with black fungus otherthan extremophilic and then draw the conclusion.

413-415 & 318-320 : Do the authors mean that black yeast generally lacks the  genetic component arrangement in biosynthetic clusters for melanin synthesis  but certain black yeast possess BGCs for all forms of melanin?

Indicate A in Figure 2A.

Authors may represent Figures as Figure 1-6 instead Figure A1 -A6

Table A2: Add the ncbi gene id/ locus for the represented genes.

Author Response

(The authors gave the same response as above.)
